# Iron Stress Affects the Growth and Differentiation of *Toxoplasma gondii*

**DOI:** 10.3390/ijms25052493

**Published:** 2024-02-21

**Authors:** Zhu Ying, Meng Yin, Zifu Zhu, Zheng Shang, Yanqun Pei, Jing Liu, Qun Liu

**Affiliations:** 1National Key Laboratory of Veterinary Public Health and Safety, College of Veterinary Medicine, China Agricultural University, Beijing 100083, China; yzyingzhu@163.com (Z.Y.); yinmeng9812@163.com (M.Y.); dwyxyzf@163.com (Z.Z.); 18797360881@163.com (Z.S.); 18349325943@163.com (Y.P.); liujingvet@cau.edu.cn (J.L.); 2Key Laboratory of Animal Epidemiology of the Ministry of Agriculture, College of Veterinary Medicine, China Agricultural University, Beijing 100083, China; 3National Animal Protozoa Laboratory, College of Veterinary Medicine, China Agricultural University, Beijing 100083, China

**Keywords:** *Toxoplasma gondii*, iron depletion, iron accumulation, bradyzoite differentiation

## Abstract

Iron is an indispensable nutrient for the survival of *Toxoplasma gondii*; however, excessive amounts can lead to toxicity. The parasite must overcome the host’s “nutritional immunity” barrier and compete with the host for iron. Since *T. gondii* can infect most nucleated cells, it encounters increased iron stress during parasitism. This study assessed the impact of iron stress, encompassing both iron depletion and iron accumulation, on the growth of *T. gondii*. Iron accumulation disrupted the redox balance of *T. gondii* while enhancing the parasite’s ability to adhere in high-iron environments. Conversely, iron depletion promoted the differentiation of tachyzoites into bradyzoites. Proteomic analysis further revealed proteins affected by iron depletion and identified the involvement of phosphotyrosyl phosphatase activator proteins in bradyzoite formation.

## 1. Introduction

*Toxoplasma gondii* is an obligate intracellular parasite capable of infecting the nucleated cells of almost all warm-blooded vertebrates [1,2]. Due to the wide range of intermediate hosts, *T. gondii* has the metabolic flexibility to adapt to different host environments and physiological changes. In intermediate hosts, *T. gondii* proliferates asexually in two forms, including tachyzoites causing acute infection with rapid proliferation, and slow-growing bradyzoites causing chronic infection [3]. Upon invasion of host cells, the parasite relies on the substantial uptake of nutrients from the host to satisfy its nutritional requirements for rapid proliferation, including lipids, proteins, and various metal ions [4,5]. Among these metal ions, iron holds particular significance as the most abundant biological metal ion [5,6]. Iron constitutes the third most abundant element in *T. gondii*, after zinc and copper, and accounts for 5% of the parasite’s total elemental content [7]. Iron plays a crucial role in several essential cellular processes during *T. gondii* infection, including iron-sulfur cluster biogenesis [8,9,10], heme biosynthesis [11,12], iron-dependent redox regulation [13,14,15], ATP production and energy transfer [15]. Furthermore, iron helps *T. gondii* resist interferon-induced growth inhibition [16]. However, even after decades of research, the strategies of iron metabolism in *T. gondii* remain unclear, including iron acquisition, storage/detoxification, trafficking, and iron-regulated protein expression [17]. To date, only the rhoptry proteins ROP4 and ROP2 are known to bind human lactoferrin [18,19] and the vacuolar iron transporter (VIT) that mediates iron detoxification [20].

Iron is an essential nutrient for all living organisms as the reversible transfer of electrons between ferrous (Fe^2+^) and ferric (Fe^3+^) iron is crucial for various enzymatic processes. However, excess iron is toxic as it contributes to oxidative damage [21,22]. Indeed, the host also employs elaborate strategies to keep iron concentrations within the optimal physiological range. The majority of iron in the body is stored in erythroid cells and is recycled from senescent erythrocytes by macrophages. Transferrin and lactoferrin bind to virtually all free iron in serum, bodily fluids, and tissues, whereas ferritin chelates intracellular-free iron ions. This mechanism, known as nutritional immunity, reduces the concentration of available iron to 10^−24^ M, thereby preventing pathogen exposure to iron [23,24]. During certain physiological processes or diseases, local iron accumulation or even iron overload may occur in the body, such as cellular senescence (up to 30-fold), inflammatory changes, ferroptosis, and neurodegenerative disorders (e.g., Parkinson’s disease, multiple sclerosis, and Alzheimer’s disease) [25,26,27,28]. In reality, iron levels in the human body are dynamically regulated, fluctuating between deficiency and overload under different physiological processes.

When *T. gondii* invades host cells, intracellular levels of iron ions are unable to meet the parasite’s nutritional demands for rapid proliferation. *T. gondii* needs to overcome the barriers of “nutritional immunity” to compete with its host and successfully obtain iron. At the same time, *T. gondii* needs to be cautious about increasing iron levels to avoid iron toxicity. Consequently, perturbing iron homeostasis is considered an attractive strategy for controlling *T. gondii* infection. Recent studies on *Plasmodium* spp. have shown that iron-deficiency anemia protects against malaria, whereas iron supplementation increases susceptibility and morbidity [29,30,31]. Unlike *Plasmodium* spp., which primarily parasitizes erythrocytes containing high iron levels in the form of hemoglobin during its asexual lifecycle [32], *T. gondii* encounters more iron stress as it parasitizes various cells throughout its lifecycle. Given *T. gondii*’s sophisticated lifestyle [33], it is adept at adapting to varying iron levels to ensure adequate iron supply and the detoxification of iron overload [34]. However, it is still unclear how *T. gondii* adapts to different iron levels in diverse environments.

Therefore, our study aims to investigate the impact of iron stress on *T. gondii*, including iron depletion and iron accumulation, as well as the parasite’s response to these stresses. There is no doubt about the importance of iron to *T. gondii*. In this work, we discovered that iron depletion and iron accumulation both affect the growth and survival of *T. gondii*. Notably, we observed that iron depletion triggers the transformation of tachyzoites to bradyzoites, whereas iron accumulation influences redox ability but enhances adhesive capacity. Further proteomic screening revealed that phosphotyrosyl phosphatase activator protein is involved in bradyzoite differentiation.

## 2. Results

### 2.1. Iron Stress Affects the Proliferation and Invasion of Toxoplasma gondii

Since iron in the body mainly exists in two forms, either as free iron ions or bound to proteins, it is difficult to experimentally distinguish between different iron levels. An iron-depletion environment was constructed by adding the permeable iron chelator deferoxamine mesylate (DFO) to the culture medium [35], while an iron accumulation environment was created using ammonium iron(II) sulfate as an iron supplement [36]. The inhibitory effect of different concentrations of DFO on the growth of *T. gondii* was determined using an RH strain expressing luciferase (RH-Luc). DFO inhibited the growth of RH-Luc in a dose-dependent manner, with a half-maximal inhibitory concentration (IC_50_) of 13.56 μM for the RH strain when the median toxic concentration (TC_50_) of HFF cells was 461.1 μM (Figure 1A and Appendix A). In the DFO-induced iron-depletion environment, the proliferation ability of *T. gondii* was significantly inhibited (Figure 1B), leading to a reduced area of plaque formation caused by *Toxoplasma* infection (Figure 1I). The escape ability of *T. gondii* was not affected by iron chelators. The invasion ability of *T. gondii* was decreased only after treatment with high concentrations of DFO (50 μM), (*p* = 0.0050, Figure 1C,D).

With increasing concentrations of iron supplements, supplementing divalent iron ions also inhibited the proliferation of *T. gondii* (IC_50_ = 312.9 μM, Figure 1E,F) and greatly reduced the area of plaque formation (Figure 1J). Exogenously added iron ions were less toxic to HFF cells and even promoted cell proliferation (Appendix A). Iron accumulation promoted the escape of *T. gondii* from the parasitophorous vacuoles (Figure 1H). Interestingly, as the concentration of exogenous divalent iron ions in the culture medium increased, the ability of *T. gondii* to invade host cells also increased (Figure 1G).

These findings suggest that growth of tachyzoites requires appropriate iron levels and that either too high or too low a concentration of iron can affect survival of *T. gondii*. The use of the impermeable iron chelator bathophenanthroline disulfonic acid (BPDS) and iron(III) chloride was also attempted, but it was found that only high concentrations of these additives could damage *T. gondii* (Appendix A).

### 2.2. Iron Stress Affects the Oxidation-Reduction Ability of Toxoplasma gondii

Considering that iron has the ability to transfer electrons reversibly between ferrous (Fe^2+^) and ferric (Fe^3+^) [21], it plays a significant role in the redox processes of organisms. A further investigation was conducted to determine whether the effect of iron stress on the growth of *T. gondii* was caused by an imbalance in iron-dependent redox capacity. By using a fluorescent probe for divalent iron ions (FerroOrange), it was observed that iron levels decreased when *T. gondii* was incubated with the iron chelator, DFO, and increased when the iron supplement, ammonium iron(II) sulfate, was added (Figure 2A). By labeling mitochondrial outer membrane protein Tom40 and apicoplast enoyl-acyl carrier protein reductase (ENR) in *T. gondii*, the study observed that varying iron levels did not affect the morphology and integrity of mitochondria and apicoplast (Appendix A), which are considered to be critical mediators of cellular redox homeostasis. Moreover, mitochondrial membrane potential remains unaffected by iron depletion and iron accumulation in *T. gondii*, as shown by the cation probe JC-1 [37], which reflects mitochondrial integrity (Figure 2D). However, upon incubation with iron supplements rather than iron chelators, *T. gondii* produced large amounts of oxidation products such as oxidized glutathione (GSSG) and malondialdehyde (MDA), a product of lipid peroxidation (Figure 2B,C). In addition, iron accumulation induced *T. gondii* to generate more reactive oxygen species (ROS) and superoxide anions (Figure 2E,F). These findings indicate that iron accumulation creates an imbalance in the redox capacity of *T. gondii*, which explains why the addition of exogenous ferrous iron hinders the proliferation of *T. gondii*.

### 2.3. Enhanced Adhesion Ability of Toxoplasma gondii in a High Iron Environment

To gain a deeper understanding of the impact of iron accumulation on *T. gondii*, RNA sequencing (RNA-seq) was performed to investigate the transcriptomic changes in *T. gondii* after 24 h of Fe^2+^ addition. Among 8470 genes, 796 were identified as differentially expressed genes (log2 fold ≥ 1 or ≤−1, *p* < 0.05), of which 674 were significantly up-regulated and 122 were down-regulated under the iron-accumulation condition (200 μM ammonium iron(II) sulfate added to the media) compared to normal media (Figure 3A, Appendix A). Notably, several SAG1-related sequence (SRS) superfamily proteins belong to the upregulated genes, which are thought to mediate attachment to host cells [38,39,40]. *T. gondii* can successfully invade host cells through the following stages: (1) gliding along the host cell surface, where parasite surface proteins interact with cell surface or substrate receptors; (2) active invasion and formation of the “moving junction” (MJ) in the area of close contact between the parasite apex and the host cell; (3) resulting in the formation of parasitophorous vacuoles [41,42,43]. To further differentiate the stages affected by iron accumulation during the invasion process, the permeabilized and non-permeabilized cells were labeled with different antibodies to distinguish between *T. gondii* adhering to the cell surface or already invading the cells. The adhesion ability of *T. gondii* was found to be significantly enhanced by the addition of divalent iron ions to the culture medium or by pre-incubation with divalent iron ions prior to invading host cells, but the invasive ability was not affected (Figure 3B).

### 2.4. Conversion of Tachyzoites to Bradyzoites Induced by Iron Depletion

Similarly, RNA-seq was employed to investigate transcriptomic changes in *T. gondii* 24 h after addition of DFO. The volcano plot demonstrated that 238 genes were significantly up-regulated and 171 genes were significantly down-regulated under iron depletion conditions (log2 fold ≥1 or ≤−1, *p* < 0.05, Figure 4A, Appendix A). Notably, several genes specifically expressed at the bradyzoite stage were identified among the up-regulated genes, including LDH2 [44,45,46,47], BRP1 [48], H2A1 [49], and MIC13 [50]. Comparing the 50 genes with the highest differential expression levels among the up-regulated genes and the 30 genes with the highest differential expression levels among the down-regulated genes with the transcriptomics of *T. gondii* at different lifecycle stages in the database (including tissue cysts, chronic infection, and merozoites) [51,52,53], the study found similarities between the transcriptomics of *T. gondii* in iron-depletion conditions and the transcriptomics observed during tissue cysts and chronic infection (Figure 4B). The PRU strain was cultured in DFO-added medium for 72 h, and we observed that nearly half of the parasitophorous vacuole stained positively using *Dolichos biflorus lectin* (DBL), although some vacuoles stained faintly with DBL (Figure 4C,E). BAG1 antibody staining showed that nearly 40% of the vacuoles converted to bradyzoites (Figure 4D,F). This suggests that tachyzoites of *T. gondii* have a propensity to differentiate into bradyzoites under iron depletion, which is a stressor that drives differentiation.

### 2.5. Comparative Proteomics Reveals Differentially Expressed Proteins of Toxoplasma gondii Impacted by Iron Depletion

The conversion of tachyzoites to bradyzoites is critical for *T. gondii* transmission, and reactivation of persistent bradyzoites is the greatest difficulty in the treatment of toxoplasmosis [54]. We are excited by the fact that iron depletion (a form of nutrient starvation) can promote bradyzoite differentiation of *T. gondii*. We hope to discover new molecular mechanisms driving bradyzoite differentiation, although the transcriptome provides too much information. The study plans to further screen proteins that play a role in the iron-depletion-driven differentiation of *T. gondii* tachyzoites to bradyzoites by comparative proteomics. The following four groups of samples were prepared for proteomics analysis, including (1) negative control group: RH strain grown in normal DMEM medium; (2) 10 μM DFO group: RH strain after three generations of continuous culture in medium supplemented with 10 μM DFO; (3) 50 μM DFO group: RH strain grown under the pressure of extremely high concentration of DFO (50 μM) after three generations of continuous culture in medium with gradually increasing DFO concentration (10 μM, 20 μM, 30 μM); (4) recovery group: transferring the RH strain from the 50 μM DFO group to normal culture medium and culturing it for one generation (Figure 5A). In fact, *T. gondii* has a hard time growing in an environment with extremely high concentrations of DFO. In order to obtain a sufficient amount of *T. gondii*, the concentration of DFO in the culture medium must be gradually increased to give the parasite a buffer time to adapt to the low-iron environment. In addition, the study found that the proliferation ability of *T. gondii* grown in medium containing high concentrations of DFO did not return to normal immediately after being transferred to normal medium (Appendix A). An irreversible effect of DFO on certain proteins of *T. gondii* was speculated to be possible. As a result, samples from the recovery group were prepared to focus on these proteins.

Standard criteria were adjusted for *p*-values < 0.05 and >2 or <0.5 fold change. Accordingly, differential expression analysis yielded 267 DEGs (148 up-regulated and 119 down-regulated) and 409 DEGs (197 up-regulated and 212 down-regulated) for the 10 μM DFO group compared to the negative control group and the 50 μM DFO group compared to the negative control group, respectively. Furthermore, 21 up-regulated and 54 down-regulated proteins were found in the recovery group compared to the negative control group (Figure 5B). Of these, five proteins were up-regulated in the three experimental groups, and three proteins were down-regulated in the three experimental groups (Figure 5C). In addition, volcano plots showed the results of DEGs for the three experimental groups (Figure 5D, Appendix A). Notably, the bradyzoite stage-specific expressed proteins BAG1 and LDH2 only appeared in the 50 μM DFO group.

A GO analysis was performed to study the biological changes under iron depletion in the cellular component (CC), biological process (BP) and molecular function (MF) categories. The results show that the top CC terms all include “membrane” and “ribosome”, “cis-Golgi network” appears in the 10 μM DFO group, and “extrinsic component of the mitochondrial inner membrane” appears in the 50 μM DFO group. The top five BP terms in the 10 μM DFO group were “protein phosphorylation”, “metal ion transport”, “organophosphate ester transport”, “pigment metabolic process” and “heme metabolic process”, while the top five BP terms in the 50 μM DFO group were “ketone biosynthetic process”, “quinone metabolic process”, “quinone biosynthetic process”, “ubiquinone metabolic process”, “ubiquinone biosynthetic process”, and “small molecule biosynthetic process”. In the MF category, “protein phosphorylation”, “metal ion transport”, “organophosphate ester transport”, “pigment metabolic process” and “heme metabolic process” were the most abundant items in the 10 μM DFO group, when “protein kinase activity”, “structural constituent of ribosome”, “transporter activity”, “transmembrane transporter activity” and “electron transfer activity” were the most abundant items in the 50 μM DFO group (Figure 5E,F, Appendix A).

To gain further insight into the metabolic pathways, a KEGG analysis was performed on the up-regulated and down-regulated genes, respectively (Figure 5G,H). Some metabolic pathways were strengthened to resist the damage caused by iron depletion, such as “pantothenate and CoA biosynthesis”, “arginine and proline metabolism”, “glucosinolate biosynthesis”, “lysine biosynthesis”, and “valine, leucine and isoleucine biosynthesis”. Concurrently, some metabolic pathways are inhibited due to iron depletion, such as “Ribosome”, “sphingolipid signaling pathway”, “phosphonate and phosphinate metabolism”, “oxidative phosphorylation”, and “apoptosis”.

### 2.6. Phosphotyrosyl Phosphatase Activator Protein Is Important for Iron Depletion-Driven Differentiation of Tachyzoites into Bradyzoites

Among the differentially expressed proteins in the recovery group that are considered to be irreversibly affected by DFO, the study focused on one protein, the phosphotyrosyl phosphatase activator (PTPA) protein (TGGT1_283720), an activator of protein phosphatase 2A (PP2A) [55], which is a key regulator of starch metabolism and bradyzoite differentiation in *T. gondii* [56]. The PTPA protein has a PPTA domain (Appendix A) and an additional domain with unknown function (Appendix A). The domain with unknown function and the N-terminal redundant randomly coiled sequences may play a special role in *T. gondii*. TgPTPA is very conserved and is in the same branch as apicomplexa protozoa in the phylogenetic tree (Appendix A). The function of the PTPA protein was further evaluated by the auxin-inducible degron (AID) system. In the RHΔku80::TIR1 strain (RH::TIR1 parental), the C-terminal end of the PTPA protein was tagged with a Ty-tag-auxin inducible degron (Ty-AID) at the endogenous locus (Figure 6A). The PTPA protein was localized in the nucleus of *T. gondii*, and was efficiently degraded upon the addition of auxin (indole-3-acetic acid, IAA), as shown by indirect immunofluorescence (Figure 6B). After conditional degradation of the PTPA protein, the proliferation of *T. gondii* was significantly inhibited and the ability to form plaques was greatly reduced, but the existence of the parasites could still be observed under the microscope (Figure 6C,D). This suggests that PTPA protein plays an important role in the growth and survival of *T. gondii*. The expression of PTPA protein varied at different iron levels. The expression level increased with the addition of DFO and decreased under iron accumulation, as shown by immunoblotting analysis (Figure 6E). To clarify the relationship between PTPA protein expression as affected by iron levels and iron depletion-driven bradyzoite differentiation, an IAA and DFO were added to the culture medium, and the conditional knockout strain was cultured for 72 h. Degradation of PTPA protein in normal culture medium only seriously hinders the proliferation of *T. gondii* and does not promote the differentiation of its tachyzoites to bradyzoites. However, degradation of PTPA protein under iron-depletion conditions affected the differentiation of bradyzoites. The number of DBL staining-positive parasitophorous vacuoles was reduced, and DBL staining appeared uneven and discontinuous (Figure 6F,G). This suggests that the PTPA protein is involved in the formation of *T. gondii* bradyzoites. Under iron depletion, *T. gondii* increases the expression of the PTPA protein and promotes its transformation into bradyzoites in response to iron stress.

The PTPA protein is an essential and highly conserved protein that stimulates the tyrosyl phosphatase activity of PP2A. It has been suggested that PTPA alters the relative specificity of PP2A from phosphoserine/phosphothreonine substrates to phosphotyrosine substrates in an ATP hydrolysis-dependent manner [55]. The PTPA protein of *Plasmodium falciparum* can play a role in the regulation of the *P. falciparum* cell cycle through its PP2A regulatory activity, and five of the six amino acids that interact with PP2A in humans are conserved in *P. falciparum*. These five of the six amino acids are also conserved in *T. gondii* and are important for the direct interaction between PTPA and PP2A (Appendix A). PP2A holoenzyme contributes to protein dephosphorylation and dephosphorylates eight phosphorylation sites of the bradyzoite formation deficient 2 (BFD2, TGME49_311100) [56]. BFD2 interacts with the BFD1 transcript under stress, and deletion of BFD2 decreases BFD1 protein levels [57]. BFD1 accumulates during stress and its synthetic expression is sufficient to drive differentiation [58]. Given that PP2A is essential for BFD2 dephosphorylation and that there is a direct interaction between PTPA and PP2A, further investigation of the PTPA-PP2A-BFD2-BFD1 connection may explain the complex mechanism of tachyzoite-to-bradyzoite differentiation under iron stress.

## 3. Discussion

Iron is an essential micronutrient for all living organisms, serving as a component of iron-containing organic compounds and playing a vital role in various iron- and heme-containing proteins involved in energy metabolism and oxygen transport [59,60]. Mammals have developed intricate iron homeostatic mechanisms to meet metabolic needs and minimize the risks associated with iron overload toxicity [21,22]. Pathogens must find ways to overcome nutritional immune barriers and acquire sufficient iron from their hosts. However, the mechanism of how *T. gondii* robs iron from the host is currently unclear. Some studies on iron acquisition by other parasites have shed some light, such as *Trypanosoma brucei*, which acquires iron from the host transferrin through specific transferrin receptor-mediated endocytosis localized in the flagellar pocket at the bloodstream stage, and *Leishmania* species, which reduces the release of ferric iron from the transferrin via the *Leishmania* ferric reductase 1 (LFR1), and the transport of ferrous iron to the cytosol via the *Leishmania* iron transporter 1 (LIT1), a member of the ZIP family [34]. Presumably, there are two possibilities for iron uptake from the host by *T. gondii*, either through the acquisition of iron-containing proteins or the direct ingestion of iron ions. Indeed, Stephen’s research found that *T. gondii* steals host ferritin heavy chain (FHC), ferritin light chain (FLC), transferrin Receptor 1 (TfR1), and transferrin (Tf) proteins during acute infection [61]. However, it remains unclear how these iron-containing proteins are recruited into the periphery of parasitophorous vacuoles and how *Toxoplasma* uptakes these proteins into the cytoplasm.

Consequently, perturbing iron homeostasis is considered an attractive strategy for controlling *T. gondii* infections. Reducing cellular iron availability by adding iron chelators has been shown to have a significant inhibitory effect on *Toxoplasma* proliferation, for example, DFO decreased parasite multiplication in both human villous (BeWo) and extravillous (HTR-8/SVneo) trophoblast cells [62]. Indeed, we believe that the chelation of intracellular free ferrous iron by DFO is very similar to the process by which host cells retain excess iron ions via ferritin. When the presence of iron ions cannot be completely eliminated, the use of permeable DFO can simulate an intracellular low-iron environment. Iron depletion-mediated bradyzoite differentiation is unexpected but understandable, as nutrient deficiency is the main driver of bradyzoite transformation. Several teams have now confirmed this conclusion in their preprint papers. Clare’s team showed that iron depletion leads to significant transcriptional changes and that iron-mediated post-transcriptional regulation exists in *T. gondii* [63]. Sébastien’s team used another iron chelator 2,2′-bipyridine (bipyridyl, BPD) to conduct their research. They found that different parasite strains have different tolerances to iron depletion, and that iron depletion causes a marked perturbation of lipid homeostasis in *T. gondii* [64]. In this study, genes that were specifically expressed in the bradyzoite stage were first identified among the differential genes in the iron-depletion transcriptome. Although a RH strain that does not differentiate into bradyzoites under normal circumstances was used, the parasitophorous vacuoles showed positive DBL staining, indicating the formation of bradyzoites after the addition of DFO to the culture. The effect of iron depletion on *Toxoplasma* protein expression was further explored via proteomics. In fact, iron depletion was found to cause greater damage to *T. gondii* than bradyzoite conversion induced by glucose starvation or cholesterol starvation [65], as iron ions play a central role in many metabolic processes, such as phosphonate and phosphinate metabolism and oxidative phosphorylation. The inhibitory effect of DFO on *T. gondii* in vitro was remarkable, but treatment of *T. gondii*-infected mice with DFO did not alter the survival rate of the animals [66]. This is despite the fact that DFO has been reported in the literature to induce a 70% protection of Swiss mice against the RH strain of *T. gondii* when injected via the intraperitoneal route during acute infection [67]. The DFO treatment experiment was repeated on acute *Toxoplasma*-infections in mice, and the death curves of *Toxoplasma*-infected mice treated with iron chelators were not statistically significant compared with those of untreated mice (Appendix A). Although iron chelation treatment reduced the number of *T. gondii* in the peritoneal fluid and lungs of the mice, it promoted dissemination and increased parasite burden in other tissues, including the kidneys, spleen, and liver. Therefore, simply reducing iron availability will not control *Toxoplasma* infections, but key proteins related to iron metabolism should be targeted.

Considering iron toxicity, iron accumulation or iron overload is not common during normal physiological processes. Since humans have no regulatable excretory pathway to eliminate excess iron after absorption, iron can easily accumulate when exogenous iron is loaded excessively as a result of genetic factors, repeated blood transfusions, and other diseased conditions. Iron accumulation has been clearly implicated in pathological conditions, even in metabolic diseases hitherto thought to be unrelated to iron, such as cirrhosis, diabetes, and heart failure [68]. Pregnant women, in particular, have a considerable increase in their body’s iron requirements during pregnancy and are advised to take iron supplements to meet their own nutritional needs and fetal growth and development [69]. In fact, acute infections during pregnancy and potential harm to the fetus and newborn are the main reasons why people are scared of *Toxoplasma*. It is unknown whether iron supplementation during pregnancy and iron accumulation involved in the progression of other chronic diseases has an impact on *Toxoplasma* infection. Our research found that iron accumulation promotes an imbalance in the redox capacity of *T. gondii*. This can be interpreted as a Fenton reaction involving iron ions leading to the accumulation of reactive oxygen species and promoting the inactivation of the intracellular antioxidant system. To our surprise, the adhesion ability of *T. gondii* was enhanced during the invasion process when the iron ion content in the environment was increased. The unexpected enhancement of adhesion ability in an iron-overloaded environment may be attributed to the up-regulation of SAS proteins known to be associated with adhesion [38]. The study also revealed that iron supplementation in *Toxoplasma*-infected mice resulted in significant weight loss and increased the area of liver inflammatory necrosis (Appendix A). Considering *Toxoplasma* preference for iron, people who are taking iron supplements or patients with chronic diseases associated with iron accumulation need to prevent *Toxoplasma* infection.

Although this study sheds light on the effects of iron stress on the growth and differentiation of *T. gondii*, it is crucial to acknowledge the limitations of our research. First, a further impact on the survival of *T. gondii* by damage to host cells from iron chelators and iron supplements cannot be ruled out. Second, the study’s attempts to screen for proteins critical for iron metabolism were limited due to the diverse biological effects of iron. Third, the contribution of PTPA proteins to differentiation was not further dissected by this study. These limitations highlight the need for future studies to be more in-depth in order to more fully understand the interactions between iron, the host, and *T. gondii*.

## 4. Materials and Methods

### 4.1. Host Cells and Parasite Culture

In this study, the RH, RH∆ku80, RH-Luc, RHΔku80::TIR1 and PRU strains (laboratory storage) were utilized. Tachyzoites of *Toxoplasma gondii* were grown in confluent monolayers of human foreskin fibroblast (HFF, maintained in our laboratory) cells cultured in Dulbecco’s modified Eagle’s medium (DMEM, M&C Gene, Beijing, China) supplemented with supplemented with 2% fetal bovine serum (FBS, Solarbio, Beijing, China), 100 mg/mL streptomycin and 100 units/mL penicillin (M&C Gene, Beijing, China) at 37 °C with 5% CO_2_. Prior to infection, HFFs were grown to confluency in DMEM supplemented with 8% FBS. Prior to the infection with confluent HFF monolayers, parasites were scraped and syringe lysed using a 22-gauge needle, filtered by filtration through 3 µm polycarbonate membranes.

### 4.2. In Vitro Inhibition Assay

A total of 1 × 10^3^ tachyzoites of RH-Luc strains were inoculated into HFFs grown in 96-well plates using DMEM containing gradient diluted iron chelating agents DFO (Topscience, Shanghai, China) or iron supplements of ammonium iron(II) sulfate (Aladdin, Shanghai, China). Dimethyl sulfoxide (DMSO, M&C Gene, Beijing, China) was used as a control. After 72 h of culture, the relative luminescence units (RLU) were measured using a fluorescence microplate reader (Tecan, Spark 10M, Männedorf, Switzerland) and Firefly Luciferase Reporter Gene Assay Kit (Beyotime Biotech, Shanghai, China). The inhibition rate was calculated using the formula: inhibition rate = [(RLUDMSO − RLUexperimental group)/RLUDMSO] × 100%. Nonlinear regression curve fitting was employed to determine the median lethal concentration (IC_50_) and 95% confidence interval (CI). Three independent experiments were performed.

### 4.3. Cytotoxicity Assay

After 5000 HFF cells were cultured in a 96-well plate for 24 h, the medium containing the gradient diluted compounds was replaced. After another 48 h of culture, CCK-8 (Topscience, Shanghai, China) was added according to the manufacturer’s instructions to determine cell viability. DMSO was added as a control. Absorbance was measured at 450 nm using a Microplate Absorbance Reader (BioRad, California, USA). Three independent experiments were performed.

### 4.4. Immunofluorescence Assays

Immunofluorescence assays (IFAs) were carried out as previously described [70]. Briefly, HFFs infected with tachyzoites were fixed with 4% formaldehyde, permeabilized with 0.25% Triton X-100 (Solarbio, Beijing, China) in phosphate-buffered saline (PBS, Solarbio, Beijing, China), and blocked with 3% bovine serum albumin (BSA, M&C Gene, Beijing, China) in PBS. Cells were then incubated with primary antibody diluted in 3% BSA/PBS for 1 h at 37 °C. After washing with PBS, the cells were incubated with secondary antibodies (FITC-conjugated goat anti-mouse IgG (H + L), 1:100, or Cy3-conjugated goat anti-rabbit IgG (H + L), 1:100, Proteintech, Wuhan, China) for 1 h at 37 °C. Finally, cells were sealed with an antifade mounting medium containing DAPI (Solarbio, Beijing, China) and images were captured using a fluorescence microscope (Olympus, Tokyo, Japan).

### 4.5. Invasion Assay

A total of 1 × 10^6^ tachyzoites were incubated with DFO (50 μM) or ammonium iron(II) sulfate (200 μM) for 1 h and then inoculated into HFFs [71]. After 1 h of invasion, cells were washed with PBS and fixed with 4% formaldehyde. Non-invaded (attached) tachyzoites were stained with rabbit anti-SAG1 (laboratory storage) prior to permeabilization, while invaded tachyzoites were stained with mouse anti-IMC1 (laboratory storage) after permeabilization with 0.25% Triton X-100. Red-labeled Cy3-conjugated goat anti-mouse IgG (H + L) was used to label all tachyzoites, whereas green-labeled FITC-conjugated goat anti-rabbit IgG (H + L) was used to label attached tachyzoites. Invasion efficiency was determined by counting the ratio of red- or green-labeled tachyzoites to host cells in several random fields under a fluorescence microscope. Three independent experiments were performed.

### 4.6. Replication Assay

A total of 1 × 10^5^ freshly released tachyzoites were inoculated in HFFs grown on coverslips [70]. After 1 h of invasion, HFFs were washed with PBS and the culture medium was replaced with DMEM containing DFO or iron supplements of ammonium iron(II) sulfate. After 24 h of incubation at 37 °C with 5% CO_2_, cells were washed with PBS, fixed with 4% formaldehyde, and stained with an anti-GAP45 antibody (laboratory storage). Tachyzoites in 100 parasitophorous vacuoles (PVs) were counted in several random fields using a fluorescence microscope.

### 4.7. Egress Assay

A total of 1 × 10^5^ tachyzoites were inoculated into HFFs and cultured for 28 h [72]. Parasites were stimulated with 5% ethanol in DMEM to induce egress. Once the parasites started to be released from the PVs, coverslips were fixed with 4% paraformaldehyde. IFA was performed using an anti-GAP45 antibody (1:300) to label the tachyzoite membrane. Egress efficiency was determined by counting the number of released and unreleased PVs in multiple random fields under a fluorescence microscope. Three independent experiments were conducted.

### 4.8. Plaque Assay

A total of 100 freshly released tachyzoites were inoculated into HFFs grown in 12-well plates [73]. After 1 h of invasion, HFFs were washed with PBS and the culture medium was replaced with medium containing DFO or ammonium iron(II) sulfate. After 7 days of culture, HFFs were washed with PBS, fixed with 4% formaldehyde, and stained with 2% crystal violet. The plaque size was analyzed by ImageJ 6.0.

### 4.9. Detection of Iron, Reactive Oxygen Species, Superoxide Anion, and Mitochondrial Membrane Potential

The concentration of iron(II) ions was measured using the fluorescence probe f FerroOrange (Dojindio, Kyushu, Japan). The level of reactive oxygen species (ROS) in T. gondii was assessed using the fluorescence probe DCFH-DA (Solarbio, Beijing, China). The concentration of superoxide anion was determined using the fluorescence probe dihydroethidium (DHE, Beyotime Biotech, Beijing, China). Changes in mitochondrial membrane potential were evaluated using the fluorescence probe JC-1 (Beyotime Biotech, Beijing, China). Freshly released tachyzoites were incubated in a culture medium containing DFO or ammonium iron(II) sulfate for 1 h. After washing with PBS, the parasites were treated with FerroOrange, DCFH-DA, DHE, or JC-1, and then resuspended in 500 μL PBS following two additional washes. Finally, the stained tachyzoites were analyzed using flow cytometry or a fluorescence microplate reader [72].

### 4.10. GSSG and MDA Determination

A total of 1 × 10^7^ tachyzoites were incubated in a culture medium containing DFO or ammonium iron(II) sulfate for 1 h. After washing with PBS, parasites were lysed by freezing in liquid nitrogen and thawing at 37 °C for three cycles. The supernatant of each sample was collected for the measurement of GSSG or MDA using a GSH and GSSG Assay Kit or a Lipid Peroxidation MDA Assay Kit, following the manufacturer’s instructions (Beyotime, Beijing, China).

### 4.11. Transcriptomics Analysis

HFF cells infected with tachyzoites were cultured in the medium supplemented with DFO (50 μM) or ammonium iron(II) sulfate (200 μM). After 24 h of incubation, the cells were lysed and the parasites were collected. The collected parasites were washed twice with precooled PBS and resuspended in 1 mL of TRIzol (Sangon Biotech, Shanghai, China). The samples were then flash-frozen in liquid nitrogen and sent to Shanghai Applied Protein Technology Co., Ltd. for transcriptome data analysis (Shanghai, China). Advanced heatmap plots were created using the OmicStudio tool on https://www.omicstudio.cn (accessed on 26 December 2023) [74].

### 4.12. Comparative Proteomics Analysis

Different groups of intracellular tachyzoites cultured in HFF cells were subjected to sequential syringe lysis and filtered through a 5-μm membrane. The samples were washed twice with ice-cold PBS, minced individually with liquid nitrogen, and sent to Shanghai Applied Protein Technology Co., Ltd. for comparative proteome analysis (Shanghai, China). Advanced heatmap plots and GO analysis were performed using the OmicStudio tools.

### 4.13. Mouse Infectivity Studies

Female BALB/c mice (Vitalriver, Beijing) were intraperitoneally injected with iron chelating agents DFO (300 mg/Kg) or iron supplements, ammonium iron(II) sulfate (100 mg/kg) or phosphate-buffered saline (PBS) as a vehicle control [70]. The injections were administered one day prior to the intraperitoneal injection of 100 RH tachyzoites, and the treatment was continued for seven days. A control group consisting of non-infected and untreated mice was included. The experimental mice were weighed daily. The mice were anesthetized (i.p.) and euthanized by cervical dislocation on day 8 post-infection. Blood samples were collected by puncturing the retro-orbital plexus and tissue samples (heart, liver, spleen, lungs, kidneys, brain, and lymph nodes) were collected, fixed in 10% buffered formalin, and processed for paraffin embedding and sectioning. DNA from different tissues was extracted and absolute quantitative detection of Toxoplasma-DNA from different tissues was performed via qPCR and 529 gene primers.

### 4.14. Sequence Analysis, Phylogenetics and Protein Structure Prediction

BLAST searches for PTPA protein-related sequences via NCBI (The National Center for Biotechnology Information, https://www.ncbi.nlm.nih.gov/, accessed on 21 October 2023), ToxoDB (https://toxodb.org, accessed on 21 October 2023), or VEupathDB (https://veupathdb.org, accessed on 21 October 2023) databases. Multiple alignments of best hit protein sequences were generated by ClustalW. Maximum likelihood phylogenies were performed by MEGA 7.0. Protein 3D structures were predicted using AlphaFold2 and Alphafold2-multimer and pictures were drawn via Pymol 2.2.3 software [75].

### 4.15. CRISPR-Cas9-Mediated Conditional Knockdown

Essential genes were conditionally knocked down using an auxin-induced degradation system as described previously [56]. Briefly, a CRISPR-Cas9 plasmid [73] targeting the 3′-UTR of the respective gene near the stop codon was co-transfected with amplicons flanked by short homology regions containing the Ty-AID and a DHFR marker into the RHΔku80::TIR1 (referred to as RH::TIR1) strain. The insertion of the Ty-AID tag was confirmed by IFA. For knockdown, parasites were treated with 500 µM 3-indoleacetic acid (IAA), whereas mock treatments included only the addition of 0.1% ethanol.

### 4.16. Western Blotting

Tachyzoites were isolated from the host cells and then washed with cold PBS before being treated with RIPA lysis buffer containing Protease and Phosphatase Inhibitor Cocktail. After incubating the lysates on ice, they were centrifuged, and the resulting supernatants were mixed with protein buffer loading, boiled, and separated on polyacrylamide gels by SDS-PAGE. The separated samples were then transferred to nitrocellulose membranes and blocked in 5% fat-free milk-Tris-buffered saline (TBS) supplemented with 0.2% Tween 20 (TBST) before incubation with primary and secondary antibodies.

### 4.17. Statistical Analysis

Graphs were created and statistically analyzed using GraphPad Prism software v8.0. All data were analyzed using unpaired two-tailed Student’s *t*-tests or two-way ANOVA tests.

## 5. Conclusions

This study demonstrates that iron is a vital nutrient for the survival of *T. gondii*. Iron deficiency affects various metabolic processes in *T. gondii*, inducing the differentiation of tachyzoites into bradyzoites, whereas iron overload affects redox ability but enhances the adhesion ability. Comparative proteomics revealed more information about differentially expressed proteins in *T. gondii* affected by iron depletion. The study further revealed that phosphotyrosyl phosphatase activator protein is important for iron depletion-driven differentiation.

## Figures and Tables

**Figure 1 ijms-25-02493-f001:**
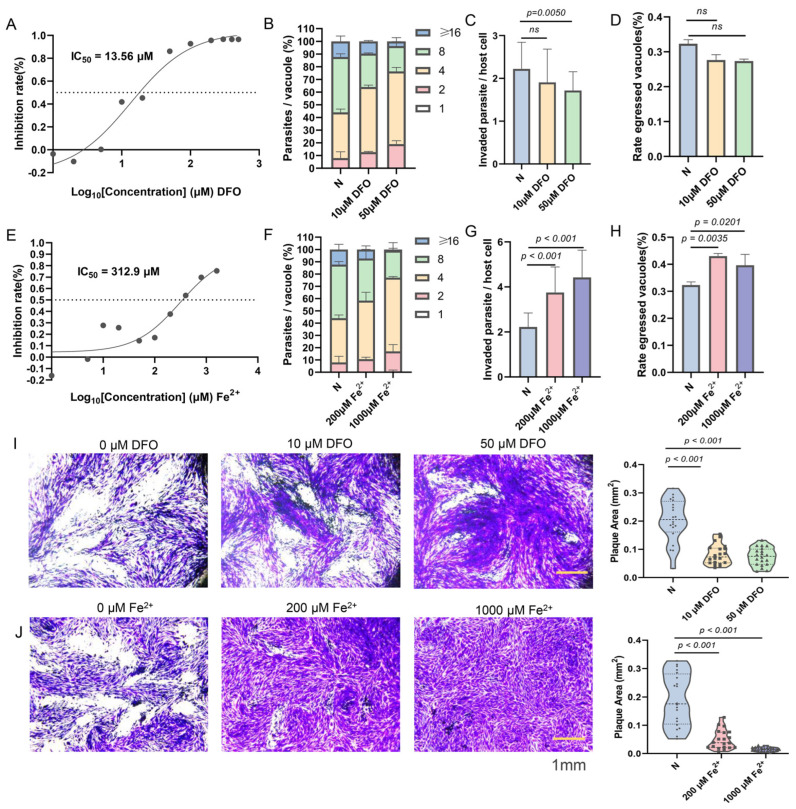
Impact of iron depletion and iron accumulation on *Toxoplasma* growth. (**A**,**E**) The half lethal concentration curve depicts the effects of different concentrations of iron chelating agents and iron supplements on *Toxoplasma* growth. RH-Luc tachyzoites were inoculated in 96-well plates containing gradient-diluted DFO or Fe^2+^ in DMEM for 72 h, and the relative light units (RLUs) were measured. Means ± SD of three independent experiments, each with three replicates, nonlinear regression-curve fitting. (**B**,**F**) The replication ability of RHΔku80 strains treated with different concentrations of iron-chelating agents (10 μM and 50 μM) and iron supplements (200 μM and 1000 μM) for 24 h. Iron depletion and iron accumulation both hinder *Toxoplasma* replication. Means ± SEM of three independent experiments, each with three replicates. A total of 100 PVs were analyzed for each replicate in each experiment. (**C**,**G**) The invasive ability of RHΔku80 strains under varying concentrations of iron-chelating agents and iron supplements. Invasive ability decreases under iron depletion but increases under iron accumulation conditions. Means ± SD of three independent experiments, each with three replicates, unpaired two-tailed Student’s *t*-test, ns: non significantly. (**D**,**H**) The escape ability of RHΔku80 strains exposed to different concentrations of iron chelating agents and iron supplements. Escape ability is enhanced under iron accumulation. Means ± SD of three independent experiments, each with three replicates, unpaired two-tailed Student’s *t*-test, ns: non-significantly. (**I**,**J**) Representative images of the plaque assays of RHΔku80 strains treated with different concentrations of iron chelating agents and iron supplements for 7 days. Images are representative of three independent experiments. Scale bar, 1 mm. Means ± SD of three independent experiments, each with three replicates, unpaired two-tailed Student’s *t*-test. More than 30 plaques were analyzed for each replicate in each experiment.

**Figure 2 ijms-25-02493-f002:**
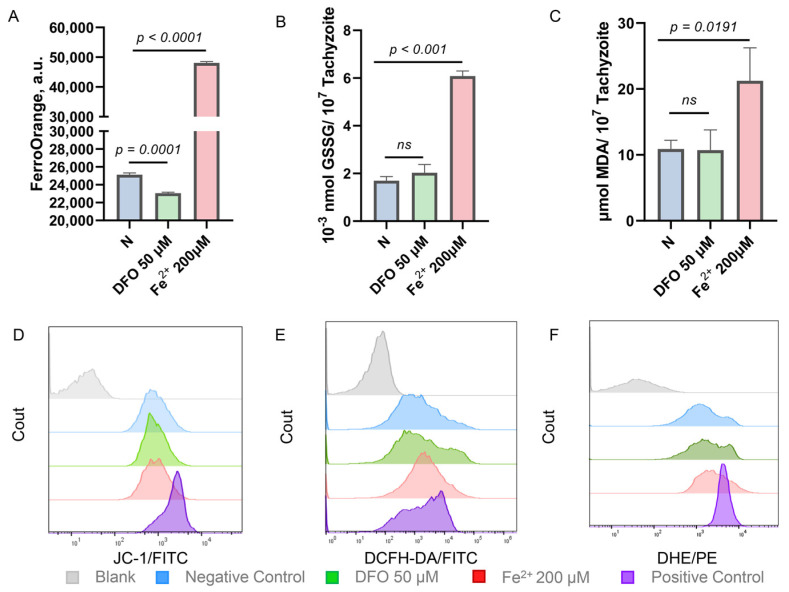
Impact of iron accumulation on *Toxoplasma* redox capacity. (**A**) The fluorescence intensity of the fluorescent probe FerroOrange reflects changes in iron levels of *T. gondii* under conditions of iron depletion and iron accumulation. Iron chelating agents reduce parasite iron levels, while iron supplements increase them. Means ± SD of three independent experiments, each with three replicates, unpaired two-tailed Student’s *t*-test. (**B**,**C**) The detection of oxidation products GSSG and MDA in *T. gondii* under different iron levels. Under iron accumulation, the content of parasite oxidation products increases. Means ± SD of three independent experiments, each with three replicates, unpaired two-tailed Student’s *t*-test, ns: non-significantly. (**D**–**F**) The analysis of changes in mitochondrial membrane potential (JC-1), ROS level (DCFH-DA), and superoxide anion concentration (DHA) in *T. gondii* under iron depletion (green) and iron accumulation (red). High iron levels increase ROS and superoxide anions but do not alter mitochondrial membrane potential.

**Figure 3 ijms-25-02493-f003:**
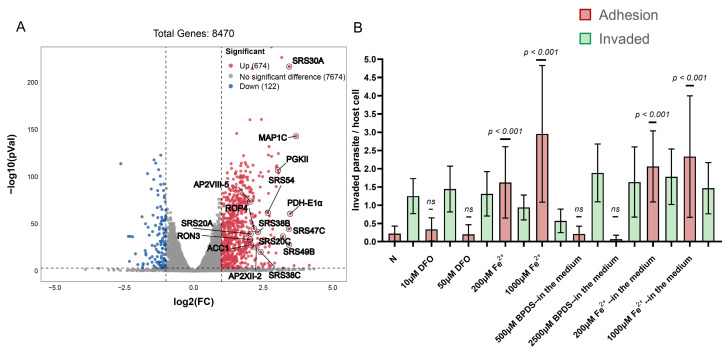
Iron accumulation affects the adhesion ability of *Toxoplasma gondii*. (**A**) The volcano scatter plot shows differential genes under iron accumulation stress through transcriptome analysis. (**B**) The difference in *Toxoplasma* adhesion ability in different iron level environments, including transfer to cells after incubating with DFO or Fe^2+^ for 1 h or adding BPDS or Fe^2+^ to the culture medium. As long as the iron ion content increases, the adhesion ability of *T. gondii* will increase. Means ± SD of three independent experiments, each with three replicates, unpaired two-tailed Student’s *t*-test, ns: non-significantly.

**Figure 4 ijms-25-02493-f004:**
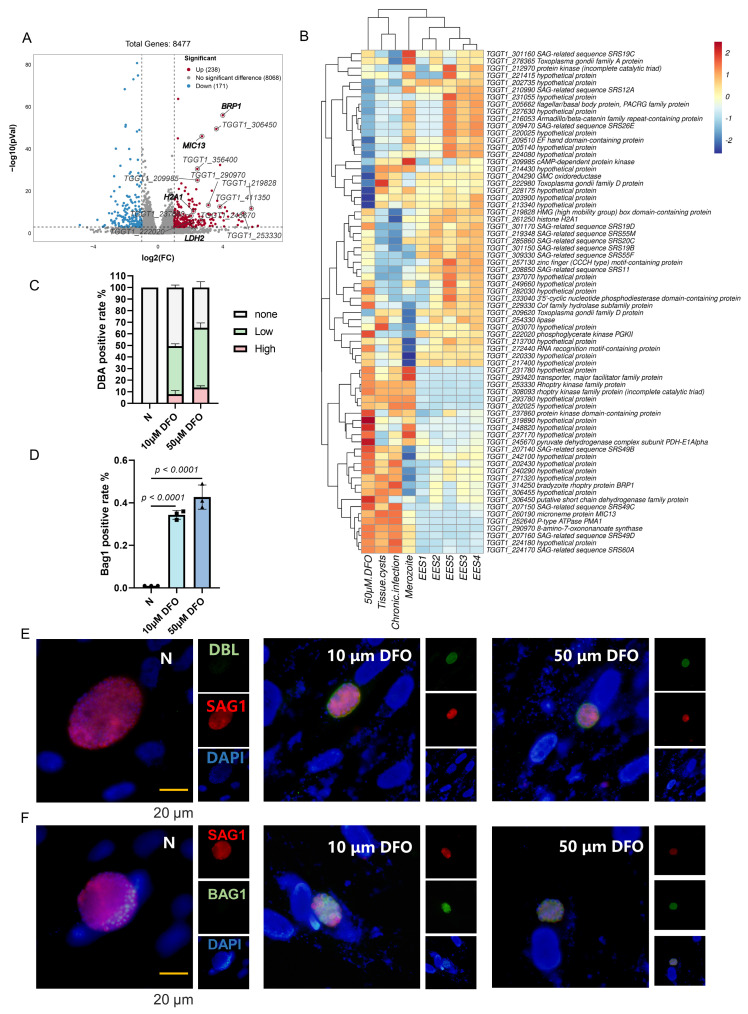
Iron depletion promotes the differentiation of tachyzoites to bradyzoites. (**A**) The volcano scatter plot demonstrats differential genes under iron depletion stress through transcriptome analysis. (**B**) The heat map comparing *Toxoplasma* transcription under iron depletion to different life stages, including tissue cysts, chronic infection, and merozoites. Similarities between the transcriptome of *T. gondii* under iron depletion and transcriptome during tissue cyst and chronic infection are observed. (**C**,**E**) Indirect immunofluorescence of Pru showing the proportion of DBL-positive parasitophorous vacuoles under iron depletion. Cells were grown in medium adding with 20 μM DFO, fixed at 72 h post infection and stained with DBL (green), anti-SAG1 (red) antibody, and Hoechst DNA-specific dye (blue). Scale bar, 20 µm. Means ± SEM of three independent experiments, each with three replicates. A total of 100 PVs were analyzed for each replicate in each experiment. (**D**,**F**) Indirect immunofluorescence of Pru showing the proportion of bradyzoite specifically expressed genes BAG1 in parasitophorous vacuoles under iron depletion. Cells were grown in medium added with 20 μM DFO, fixed at 72 h post infection and stained with anti-BAG1 (green), anti-SAG1 (red) antibody, and Hoechst DNA-specific dye (blue). Scale bar, 20 µm. Means ± SD of three independent experiments, each with three replicates. A total of 100 PVs were analyzed for each replicate in each experiment.

**Figure 5 ijms-25-02493-f005:**
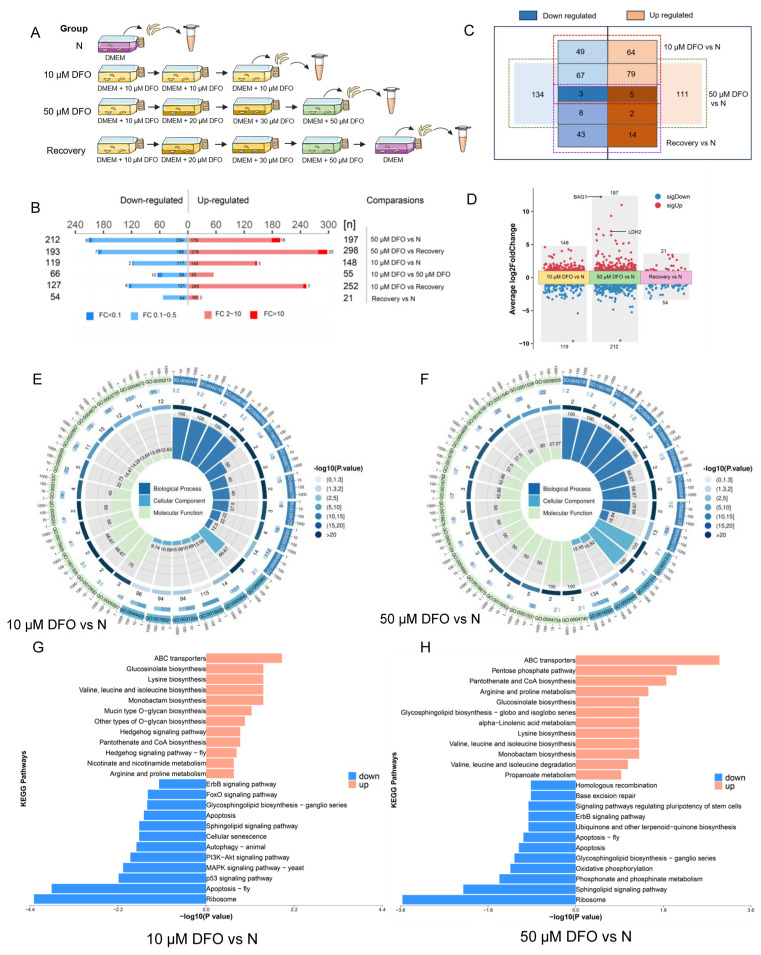
Comparative proteomic analysis of *Toxoplasma gondii* under iron depletion. (**A**) Schematic representation of sample preparation for comparative proteomics of different groups of *T. gondii*. We set up four groups of samples for proteomics analysis as follows: (1) negative control group: RH strain grown in normal DMEM medium; (2) 10 μM DFO group: RH strain after three generations of continuous culture in medium supplemented with 10 μM DFO; (3) 50 μM DFO group: RH strain grown under the pressure of extremely high concentration of DFO (50 μM) after three generations of continuous culture in medium with gradually increasing DFO concentration (10 μM, 20 μM, 30 μM); (4) recovery group: transferring the RH strain from the 50 μM DFO group to normal culture medium and culturing it for one generation. (**B**) Histogram shows the number of up-regulated and down-regulated differentially expressed proteins between different groups. Proteins with fold changes ≥2.0 or ≤−2.0 with *p* < 0.05 are highlighted in red and blue color, respectively. (**C**) Volcano plot shows the number of overlapping proteins between up-regulated and down-regulated differentially expressed proteins between different groups. (**D**) Volcano plot shows differentially expressed proteins in 10 μM DFO group vs. Negative control group, 50 μM DFO group vs. Negative control group and Recovery group vs. Negative control group. (**E**,**F**) Circle plot of the GO functional annotation targets the biological process, cell component and molecular function in 10 μM DFO group vs. Negative control group and 50 μM DFO group vs. negative control group. (**G**,**H**) Bar plot shows gene KEGG enrichment results of differentially expressed proteins in 10 μM DFO group vs. negative control group and 50 μM DFO group vs. negative control group. The X-axis labels indicate a *p*-value, and the Y-axis denotes the names of KEGG terms.

**Figure 6 ijms-25-02493-f006:**
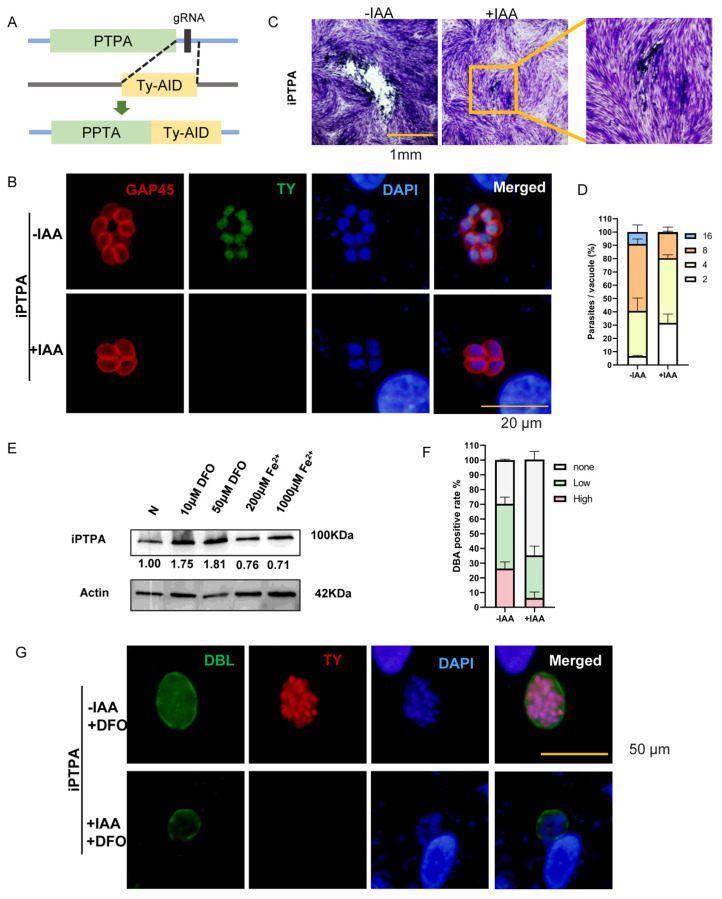
Phosphotyrosyl phosphatase activator protein involved in iron depletion-mediated bradyzoite differentiation. (**A**) The strategy used to construct the conditional knockdown strain PPTA-Ty-AID (called iPTPA for short). The AID construct (in frame with a Ty-tag) was fused to the C-terminus of endogenous PTPA by CRISPR/Cas9-mediated homologous recombination. (**B**) Depletion of PTPA expression in the iPTPA strain by IAA treatment. Intracellular parasites were treated with or without 500 µM IAA and subsequently subjected to IFA analyses using mouse anti-Ty (green) and rabbit anti-GAP45 (red) antibodies. PTPA is localized in the nucleus of *T. gondii*. Scale bar, 20 µm. (**C**) Arrest of parasite growth after PTPA depletion, as determined by plaque assays with or without IAA treatment for 7 days. Scale bar, 1 mm. (**D**) Proliferation assays on iPTPA strains treated with or without IAA for 24 h (with 22 h of ±IAA pretreatment before the proliferation assay). Means ± SEM of three independent experiments, each with three replicates. A total of 100 PVs were analyzed for each replicate in each experiment. (**E**) Expression changes of PTPA under iron depletion and iron accumulation through Western blot. Iron chelators and iron supplements were added to the cells 12 h after the worms infected the cells, and the worms were collected after another 24 h of culture to make samples. Adding iron chelators (10 μM and 50μM DFO) and iron supplements (200 μM and 1000 μM Fe^2+^) after iPTPA strains infect the cells for 12 h, samples were made after culturing for another 24 h. (**F**,**G**) Indirect immunofluorescence of iPTPA strains showing the proportion of DBL-positive parasitophorous vacuoles under iron depletion. Cells were grown in medium added with 20 μM DFO, fixed at 72 h post infection (with 70 h of ±IAA pretreatment) and stained with DBL (green), anti-Ty (red) antibody, and Hoechst DNA-specific dye (blue). Scale bar, 50 µm. Means ± SEM of three independent experiments, each with three replicates. A total of 100 PVs were analyzed for each replicate in each experiment.

## Data Availability

The original contributions presented in the study are included in the article/Appendix A, further inquiries can be directed to the corresponding author/s.

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
