# Peer review of "Iron Stress Affects the Growth and Differentiation of Toxoplasma gondii"

_ijms, 2024, doi:10.3390/ijms25052493_

Round 1

Reviewer 1 Report

Comments and Suggestions for Authors

This study assessed the impact of iron stress (iron depletion and iron accumulation), on the growth of Toxoplasma gondii. The topic is relevant in the field because authors gave the new scientific information.  Iron accumulation resulted in the disruption of the redox balance of T. gondii, and also the parasite's ability to adhere in high-iron environments was higher. On the other side, iron depletion resulted in the differentiation of tachyzoites into bradyzoites. Further proteomic analysis found the proteins affected by iron depletion and identified the involvement of phosphotyrosyl phosphatase activator proteins in bradyzoite formation. The authors cited the new relevant literature in that field and discussed their findings with other results from the literature on other parasites.

I don't have any suggestions or corrections. It is a really interesting paper with new information and contributions to scientific research on Toxoplasma gondii.

Author Response

We greatly appreciate your time and effort in reading our manuscript and providing your critical review. We are grateful for your approval of our research on the impact of iron stress (iron depletion and iron accumulation) on the growth of Toxoplasma gondii. Our study confirmed the importance of iron in the survival and differentiation of Toxoplasma gondii, and in fact follow-up research is still ongoing.

Reviewer 2 Report

Comments and Suggestions for Authors

Reviewer's report:

 It was a pleasure to review this paper investigating the role of iron stress in Toxoplasma gondii, including iron depletion and iron accumulation, as well as the parasite's response to these stresses. The rationale for this work is based on the fact that even after decades of research, the iron metabolism in T. gondii remains unclear, encompassing iron acquisition, storage/detoxification, trafficking, and iron-regulated protein expression.  The inclusion of proteomic analyses adds depth to the study, allowing for a molecular understanding of the observed effects. The experiments were conducted carefully, with a sufficient number of repetitions, yielding high-quality data and appropriate statistical analyses.

 Here are some minor considerations:

 Aim of the Study:

I would suggest authors to consider rephrasing the sentences on lines 74 to 76. For example: "We discovered that T. gondii requires an appropriate ……. " These sentences are more coherent with the results of this study, and not the aim. Please rephrase them to align with the objectives of the current study.

 Results:

Overall, the results are well-presented, with the authors providing a detailed account of the experimental design and outcomes. Please consider adding a sentence at the end of the all Figures to describe the statistical method together with p-value that supports your results;  the number of independent experiments on which the data are based (in places where it is possible). 

 Additionally, look out for technical errors on lines 278 and 318.

 Discussion:

Please, consider the following point for potential improvement of Discussion Section:

The rationale of investigation of PTPA is not clearly explained. Although the results are presented in Figure 6, the study's limitations section states: "Third, the contribution of PTPA proteins to differentiation was not further dissected by the study." Please, clarify why PTPA is not given sufficient attention in the discussion.

Author Response

Sincere thanks should be given to the reviewer and editor for the constructive comments and suggestions. The responses to the comments are given below. Here are some minor considerations:

1.Aim of the Study:I would suggest authors to consider rephrasing the sentences on lines 74 to 76. For example: "We discovered that T. gondii requires an appropriate ……. " These sentences are more coherent with the results of this study, and not the aim. Please rephrase them to align with the objectives of the current study.

Reply:

We gratefully thank you for your time spend making constructive remarks and useful suggestion. We totally understand your concern about this paragraph. We rephrase this section to describe the purpose and results of this study on lines 74 to 77. We hope this content change will satisfy you.

 2.Results:Overall, the results are well-presented, with the authors providing a detailed account of the experimental design and outcomes. Please consider adding a sentence at the end of the all Figures to describe the statistical method together with p-value that supports your results;  the number of independent experiments on which the data are based (in places where it is possible). 

 Additionally, look out for technical errors on lines 278 and 318.

Reply:

Thank you so much for your careful check. We add a sentence at the end of the each figure to describe the statistical method and the number of independent experiments on which the data are based. The p-values are already presented in the figures and therefore are not described again in the legend. And the technical errors on lines 278 and 318 are corrected.

3.Discussion:

Please, consider the following point for potential improvement of Discussion Section:

The rationale of investigation of PTPA is not clearly explained. Although the results are presented in Figure 6, the study's limitations section states: "Third, the contribution of PTPA proteins to differentiation was not further dissected by the study." Please, clarify why PTPA is not given sufficient attention in the discussion.

Reply: Thank you for your rigorous consideration. We found that degradation of PTPA protein under iron-depletion conditions affected the differentiation of bradyzoites, so we speculated that the PTPA protein is involved in the formation of T.gondii bradyzoites. But we don't know how PTPA protein affects bradyzoite differentiation. We speculate that there is some connection between PTPA, PP2A, BFD1 and BFD2, and the corresponding analysis and discussion are discussed in 2.6, lines 263 to 278. In fact, our team is conducting further research on the PTPA-PP2A-BFD2-BFD1 connection, and we hope there will be a good result to explain the complex mechanism of tachyzoite-to-bradyzoite differentiation under iron stress.